# Pain in Ehlers–Danlos Syndrome: A Non-Diagnostic Disabling Symptom?

**DOI:** 10.3390/healthcare11070936

**Published:** 2023-03-24

**Authors:** Viviana Guerrieri, Alberto Polizzi, Laura Caliogna, Alice Maria Brancato, Alessandra Bassotti, Camilla Torriani, Eugenio Jannelli, Mario Mosconi, Federico Alberto Grassi, Gianluigi Pasta

**Affiliations:** 1Department of Othopaedics and Traumatology, Fondazione Policlinico IRCCS San Matteo, University of Pavia, 27100 Pavia, Italy; 2Department of Othopaedics and Traumatology, Fondazione Poliambulanza Istituto Ospedaliero, 25124 Brescia, Italy; 3Regional Center of Ehlers-Danlos Syndrome, IRCCS Ca’Granda Foundation Ospedale Maggiore Policlinico, 20122 Milan, Italy

**Keywords:** Ehlers–Danlos syndrome, chronic pain, neuropathic pain

## Abstract

Background: Ehlers–Danlos syndrome (EDS) is a phenotypically and genetically heterogeneous group of connective tissue disorders. Currently, diagnosis of EDS is based on a series of clinical and genetic tools. On the other hand, the hypermobile form has not yet been characterized from a genetic point of view: it is considered a part of a continuous spectrum of phenotypes, ranging from isolated non syndromic joint hypermobility, through to the recently defined hypermobility spectrum disorders (HSD). The aim of this study is to characterize the pain symptom that is not considered among the diagnostic criteria but is relevant to what concerns the quality of life of patients with EDS. (2) Methods: A review of the literature was performed on two medical electronic databases (PubMed and Embase) on 20 December 2022. Study selection and data extraction were achieved independently by two authors and the following inclusion criteria were determined a priori: published in the English language and published between 2000 and 2022. (3) Results: There were fifty eligible studies obtained at the end of the search and screen process. Pain is one of the most common symptoms found in Ehlers–Danlos (ED) patients. Different causes seem to be recognized in different phases of the syndrome. (4) Conclusions: Pain is a nonspecific symptom and cannot be considered among the diagnostic criteria, but it is a negative predictive factor in the quality of life of patients with EDS. Therefore, proper evaluation and treatment is mandatory.

## 1. Introduction

Ehlers–Danlos syndrome (EDS) is a phenotypically and genetically heterogeneous group of connective tissue disorders with multiorgan involvement. Its prevalence is about 1:5000 with classical (cEDS), vascular (vEDS) and molecularly unsolved hypermobile (hEDS) variants accounting for more than 90% of all ED cases [1]. In 1970, Beighton described five types of EDS: I: severe, II: moderate, III: hypermobile, IV: ecchymosis and V: X linked [2]. Different unsuccessful classification systems were proposed, always with a poor description of clinical and genetic findings, until May 2016. Here, during the International Ehlers–Danlos conference in New York, a new classification system was proposed, with 13 types of EDS, almost each one with clinical, biochemical and molecular differences [2]. The diagnosis of ED syndrome can nowadays rely on a series of clinical and genetic tools, as recently highlighted by a published article [3]. Despite this, the diagnosis of this condition remains a challenge, even more so if one takes into consideration the hypermobile form, which has not yet been characterized from a genetic point of view. EDS diagnosis (the classical form, cEDS) is characterized by skin involvement (i.e., skin hyperextensibility and atrophic scarring) and generalized joint hypermobility. These major criteria are flanked by minor criteria, such as easy bruising; soft, doughy skin; skin fragility (or traumatic splitting); molluscoid pseudotumor; subcutaneous spheroids; hernia (or history thereof); epicanthal folds; complications of joint hypermobility (e.g., sprains, luxation/subluxation, pain, flexible flatfoot); and family history of a first degree relative who meets clinical criteria. The clinical diagnosis of cEDS requires the simultaneous presence of the two major criterion or the presence of one of the major criteria plus at least three minor criteria. After that, confirmatory molecular testing is mandatory to reach a final diagnosis [2]. As has already been said, hypermobile EDS (hEDS) is the only type of EDS form without a known molecular basis. It is characterized by generalized joint hypermobility, chronic pain, dizziness, fatigue and minor skin changes, all features that can change according to patient age and gender [4]. Nowadays, hEDS is considered as a part of a continuous spectrum of phenotypes, ranging from isolated non syndromic joint hypermobility, through to the recently defined hypermobility spectrum disorders (HSD). The clinical diagnosis requires the simultaneous presence of three reported criteria [2]: 1. generalized joint hypermobility (GJH); 2. evidence of syndromic features, musculoskeletal complications and/or family history; and 3. exclusion of alternative diagnoses. In addition to the diagnostic criteria, EDS patients can present a series of symptoms and signs, which, although not diagnostic, due to poor specificity, can become the most debilitating aspect of this syndrome. Among these, we remember sleep disturbance, fatigue, pain, postural orthostatic tachycardia, functional gastrointestinal disorders, dysautonomia, anxiety and depression.

Moreover, the symptom of pain also appears to be widespread in the EDS patient population. With the present review, we wanted to focus on pain as a symptom; it is not included in the diagnostic criteria, but it is important to consider the fact that pain represents the main cause of access to medical care in the ED patient, representing a factor of both physical and emotional disability in various age groups.

## 2. Materials and Methods

### 2.1. Identifying the Research Question

The research question identified for the literature review was first of all to highlight pain, one of the most common symptoms found in ED patients, as an undiagnosed disabling symptom. Secondly, we also intend to investigate the characteristics and the predictors of the pain to identify the most effective forms of treatment.

### 2.2. Identifying Relevant Studies

A literature search was conducted in order to find relevant studies on the topic. These ones were identified by means of Pubmed and Embase databases.

#### 2.2.1. Electronic Database Search

The following electronic databases were searched, taking into consideration the chronological span between 2000 and 2022:-PubMed-Embase

The research strategy was carefully designed in order to retrieve the most relevant results. Due to the specificity of the two databases employed, a different search string was built for each one (1: PubMed search string; 2: Embase search string). In brackets, the number of the results is given.

(“ehlers-danlos”[All Fields] AND (“pain”[MeSH Terms] OR “pain”[All Fields])) AND ((humans[Filter]) AND (1/1/2000:20/12/2022pdat]) AND (english[Filter])) [381];‘ehlers danlos’ AND ‘pain’/exp AND [01-01-2000]/sd NOT [20-12-2022]/sd AND [humans]/lim AND [english]/lim [1315].

#### 2.2.2. Other Sources

Twenty-four studies were also included, starting from a review excluded from the analysis, since only primary studies were taken into account according to the inclusion criteria. Article details are reported in Figure 1.

### 2.3. Study Inclusion Criteria

Starting from the research question, inclusion and exclusion criteria for the objective selection of the studies identified were defined as follows. Only studies published in the English language between 2000 and 2022 were eligible for inclusion.

### 2.4. Data Extraction

A standardized data extraction sheet was prepared, where the main information of the studies was collected (e.g., first author’s name, study title, publication year and DOI).

Titles and abstracts and full texts were screened by the research team—i.e., two authors performed the study selection and the data extraction independently, and all disagreement was discussed between the authors.

### 2.5. Study Selection

Via the literature search, fifty studies were included in this literature review, twenty-six of them were identified via database searches and twenty-four via citation searching (Figure 1). Figure 1 shows the process of study selection in detail, covering the number of search records retrieved from the two database searches (n = 1696) and all other searches (n = 25); the number of screened titles/abstracts (n = 1510); and the number of studies finally included (n = 50).

## 3. Results

Pain, usually assessed by the Beighton score, is one of the most common symptoms found in ED patients and seems to be inversely correlated to generalized joint hypermobility. In fact, approximately 30% of children with diagnosed hEDS reported arthralgias, back pain and myalgias, while this rate became more than 80% if considering patients over the age of 40 years, who often show a “negative” Beighton score but increased pain [4]. Sacheti et al. reported that 100% of 28 studied hEDS patients suffered from pain, with a mean score on the numeric pain rating score (NPRS) of 8 out of 10 for all types of EDS [5]. Similar values were found by Voermans et al., with 90% of all 273 studied EDS patients and 93% of them who reported joint hypermobility, who even showed the highest scores for severity of pain [6].

### 3.1. Pain Characteristics

Pain may be localized or widespread and may be acute or chronic. It can interfere with daily life and affect sleep quality with physical and emotional functional impairment [6]. It is frequently localized in the neck, shoulders, hips, forearms and legs. In particular, neck pain is frequently associated with headache, such as migraines or tension-type headache [6]. At the beginning, pain seems to be limited to a few joints and/or muscles and shows a migratory pattern. It then becomes more persistent with a generalized distribution. Patients describes pain as a burning sensation, peripheral paresthesia, generalized hyperalgesia, allodynia and hypersensitivity to different stimuli [7,8,9,10]. Even gastrointestinal, genito-urinary and pelvic areas are affected by pain [5,10,11,12,13,14,15,16,17]. The genesis of the pain is not clear. It seems to be related to nociceptive and neuropathic pain, with a component of pain sensitization. Nociceptive pain is found during early stages of acute and localized pain, and it is related to affected ligaments and tendons, joint muscles and connective tissue. Ligamentous and tendinous damage, which derive from joint instability, and the presence of microtrauma on the joint surface, which in turns lead to adaptations and overload in other areas, contribute to nociceptive pain [18,19]. Other factors contributing to nociceptive pain are reduced bone mass [20,21,22], multiple surgical procedures [23], often in patients with misdiagnosis, and premature osteoarthritis [24]. Moreover, there was even a neuropathic component for hEDS pain, as reported by Voermans and colleagues who found compressions and axonal neuropathies in different EDS subtypes [25]. In fact, this kind of pain can be explained by nerve (sub)luxations that are found to be more frequent in the upper limb area and can explain neuropathies such as paresthesia [7]. In addition, a decreased intraepidermal nerve fiber density was demonstrated in skin biopsies in these patients, which contributes to explain nerve neuropathies [26,27]. The last contributor to chronic pain seems to derive from central sensitization, which was found in many patients, suggesting a generalized hyperalgesia in patients suffering from these conditions [9,10]. This was confirmed by a research group who showed that hEDS/joint hypermobility syndrome (JHS) increased due to repeated stimuli and decreased exercise-induced analgesia. Central sensitizations could be considered because of the continuous stimulation of peripheral nociceptors by mediators released by aberrant extracellular matrix (ECM). Another factor related to chronic pain in hEDS seems to be the loss of proprioception [19,28,29,30,31], which helps maintaining balance and coordinate normal activities. To explain the loss of proprioception, different hypotheses were formulated, such as that excessive joint mobility may damage proprioceptive receptors in the joints or that the sensation of pain in the joint may diminish proprioception [29,32]. Moreover, a program in order to enhance proprioception demonstrated an improvement in pain [19]. Even muscle weakness seems to contribute to generation of chronic pain and it is in part due to the fact that people suffering pain learn to avoid behavior which in turn can cause pain and this contributes to muscle disuse and deconditioning, increasing functional disabilities [33]. Another contribution to generalized pain and its chronicization seems to come from modifications to ECM, which are demonstrated to have a key role in central nervous system (CNS) neuroplasticity and connectivity [31], and, moreover, painful stimuli can alter ECM at chronic and acute time points after injury [34,35]. Until recently, it was believed that hEDS patients’ pain was due to an alteration in small fiber function [27]. Leone et al. compared 22 hEDS patients to a group of healthy controls: the patient group reported that pain simultaneously affected axial, upper and lower segments on the right and left sides, which demonstrated that hEDS pain does not reflect joint problems. They reported that small fiber function was preserved (which was demonstrated using QST, showing high specificity and sensibility for diagnosing small fiber neuropathy) but patients showed a deficit of descending inhibitory pain control, with a “pro-nociceptive” pattern of pain modulation [36], which, in turn, can explain why drugs restoring endogenous pain inhibitory control, such as antidepressants, might be effective for reducing pain in this group of patients.

In conclusion, pain seems to have different causes in different phases of the syndrome: initially, it could be due to joint hypermobility complications, and then the persistent nociceptive input may causes central sensitization in the dorsal horn neurons and abnormalities of the endogenous pain inhibitory control [37] (Table 1).

### 3.2. Predictors of Pain

Recently, Kalisch et al. studied 75 HED patients, using different valuation scales, to find out factors that can be considered predictors of pain and mobility impairment. All participants reported pain, according to previous studies, with 42.7% suffering from severe pain. From this study, it emerged that professional status, delayed diagnosis and helplessness were predictors of severe pain, while body mass index, age and fatigue predicted mobility disability. Although physical therapies are one of the most effective tools in getting quality of life better, Bovet et al. reported that 38% of iatrogenic injuries resulted from an inadequate rehabilitative protocol, while another 30% of patients reported always feeling an increased risk of injury during a rehabilitative program. The reason seems to be the strict adherence of physical therapists to standard treatment programs without any adaptations to patient’s need [38]. One of the most common quality of life assessment scores is SF-36, used worldwide to investigate physical and emotional components [11,12,38,39,40,41]. The score ranges from 0 to 100, with lower scores suggesting worse health-related quality of life. It is made up of 36 questions, examining physical role limitations, bodily pain, general health and vitality, and mental health (Table 2).

### 3.3. Management of Pain in ED Patients

As seen so far, pain is not only one of the most frequent symptoms in ED patients but also the most frequent reason why this category of patients tends to go to a specialist in the hope of improving their quality of life. Recognizing the different origins of pain, it will be essential to act on the different components, both neuropathic and nociceptive, both from the point of view of purely pharmacological treatment and from the point of view of physical therapies and emotional/psychological support. Pharmacological treatment, in fact, is part of a much more complex therapeutic scenario, in which different forms of treatment are articulated to deal with a pain that often proves not to be responsive to a “basic” drug therapy. In this regard, unconventional treatments can represent an additional prevention and therapy tool, capable of addressing not only the purely physical aspect, but also giving support to the psychological one and teaching the patient “pain management”. These methods include psychotherapy, cognitive behavioral therapy, meditation, hypnosis and psychomotor techniques. A great help in the treatment of this pathology comes from physical therapies, such as exercises to improve posture and proprioception, muscle strengthening and physical conditioning [19]. Further help in pain management can also derive from the use of devices (orthoses, braces, etc.) prescribed by a specialist in the sector (Table 2).

## 4. Discussion

Pain may be localized or widespread and may be acute or chronic. Nociceptive pain is related to ligaments, tendons, joints, muscles and connective tissue and is present in the early stages of acute, localized pain. The main neuropathy of the nerves contributing to chronic pain appears to be central sensitization, which was observed in many patients and suggests a correlation with global hyperalgesia.

Chronic pain strictly and negatively affects ED patients’ quality of life, causing physical and emotional impairment and subsequent substantial disability. Many authors reported lower health-related quality of life scores in ED patients if compared to the healthy population [42,43]. Physical and emotional impairment is so important that results from these kind of studies put children with JHS into the same cluster of children affected by other chronic conditions such as cancer, hypoplasia of the left ventricle and obesity, in order to stress the importance of disability coming from this condition [44]. The severe psychological and social impact of these manifestations can result in anxiety and depression, with a high rate of suicidal attempts [45]. A self-reported questionnaire showed higher levels of anxiety and depression in individuals with EDS, with a correlation between fears and anxiety and hypermobility of joints [46,47], even without any psychiatric disorders [48]. Particular attention must be paid to the diagnosis of this pathology whose symptoms, often vague and poorly defined, create confusion, diagnostic delay and therapeutic errors that do nothing but worsen the health condition of these subjects, with both physical and psychological implications. The diagnostic and therapeutic approach, in fact, requires the participation of an expert multidisciplinary team who hears the alarm bells and implements all those pharmacological and non-pharmacological strategies, useful for the treatment of pain and the management of fatigue, altered proprioception, and all those debilitating alterations that the ED patient complains of. The multidisciplinary approach to the patient must be simultaneous. Isolated intervention with drug therapy alone or, conversely, with nonpharmacological therapy alone should be avoided.

In this review, we tried to classify the principal EDS pain types by reading and looking for data that are currently in the literature to better understand possible causes, the stage of onset and possible treatments (Table 1). Then, we identified three main types of pain which manifests at different times of the pathology (in early and medium–late stage). They have also different causes which lead to different therapeutic approaches, as represented in Figure 2 below.

Physiotherapy is a well-established approach that provides immediate and long-term results [50]. Physical therapy, now recognized as an integral part of pain treatment, following personalized programs based on the symptoms most frequently complained of by the subject, could prove to be an even more effective tool because it is free of side effects and complications

Psychotherapy appears to be useful in all patients. In particular, the best results were obtained in patients in whom pain negatively affects quality of life and is unresponsive to drug therapy and physiotherapy [50].

A new front of development could be that of subjecting ED patients to pain questionnaires, trying to understand in a case-specific way which is the neuropathy component, and which is the nociceptive component of the perceived pain, to propose pharmacological therapies and not aim at specific treatment of the algic component.

## 5. Conclusions

Chronic pain is the most frequent symptom, affecting 80% of ED patients older than 40 years of age. Selected studies have highlighted the negative implications of the pain symptom in EDS patients. The psychological consequences of this symptom can result in severe anxiety states and depressive syndromes.

The attention of the multidisciplinary team that takes care of the patient, therefore, must not be aimed only at the treatment of those that are recognized as the most striking and typical symptoms of the syndrome, but must be aimed at the treatment and prevention of those symptoms often considered “minor”, because they are extremely unspecific. Pain represents, among these, the most disabling symptom and its treatment requires the involvement of an experienced multidisciplinary team of experts.

The research and the evaluation of the articles has not been easy because there is not much material in the literature published in the last 10 years. We are aware that this represents a limitation of the study and consequently further research will be needed. Moreover, we used the databases most commonly accessed by clinicians to explore the research question. The choice of literature from two sources could represent a limitation of sample constitution. 

## Figures and Tables

**Figure 1 healthcare-11-00936-f001:**
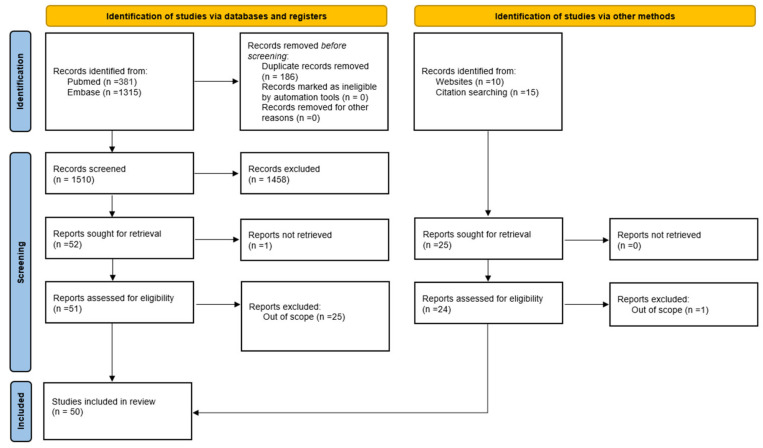
Study selection process.

**Figure 2 healthcare-11-00936-f002:**
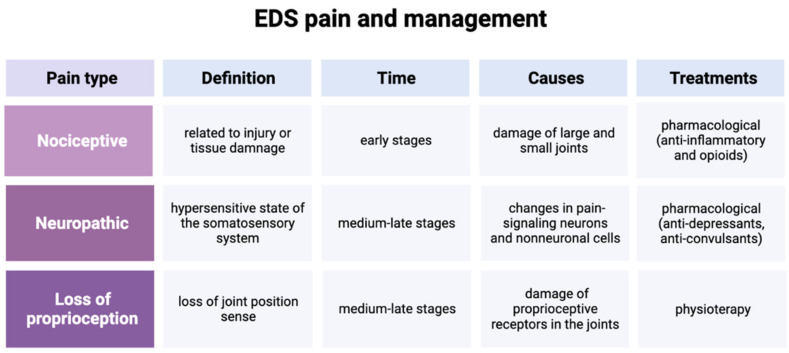
Correlation of EDS pain and management [49,50].

**Table 1 healthcare-11-00936-t001:** The characteristics of pain in various forms of EDS.

Publication	Classification of the Diseases	Patients Number	Pain Characteristics	Type of Study
Article	Classic EDS Vascular EDS TNX-deficient EDS HT-EDS	10 patients 10 patients 10 patients 10 patients	Neuropathic pain Nociceptive pain	Experimental study
Article	HT-EDS	15 patients	Neuropathic pain	Case-control
Letter	Classic EDS HT-EDS	15 patients 29 patients	Neuropathic pain	NA
Article	HT-EDS	23 patients	Neuropathic pain Nociceptive pain	Case-control
Article	Classic EDS HT-EDS	74 patients 74 patients	Chronic pain and hyperalgesia	Case-control
Article	HT-EDS	21 patients	Chronic joint/limb pain	Experimental study
Article	HT-EDS	32 patients	Joint pain and chronic pain	Case-control
Article	Hypermobility syndrome *	28 patients	Joint pain	Case-control
Review	Classic EDS Vascular EDS HT-EDS	NA	Fibromyalgia/musculoskeletal pain Chronic and neuropathic pain	NA
Article	Hypermobility syndrome *	641 patients	Chronic widespread pain	Case-control
Article	EDS	46 patients	Nociceptive pain	Case-control
Review	Hypermobility syndrome *	82 patients	Chronic joint/limb pain	NA
Review	HT-EDS	200 patients	Nociceptive pain, neuropathic limb pain, dysfunctional pain, musculoskeletal pain and visceral pain	NA
Article	Hypermobility syndrome *	18 patients	Musculoskeletal pain, joint pain	Experimental study
Article	EDS	23 patients	Nociceptive pain	Case-control
Article	Hypermobility syndrome *	25 patients	Nociceptive pain	Experimental study
Article	Hypermobility syndrome *	25 patients	Nociceptive pain	Case-control
Article	Classic EDS Vascular EDS HT-EDS EDS	45 patients 11 patients 162 patients 55 patients	Musculoskeletal pain Chronic pain	Experimental study
Article	Osteoarthritis *	384 patients	Nociceptive pain	Experimental study
Letter	Classic EDS HT-EDS	22 patients 22 patients	Chronic and neuropathic pain Moderate pain for at least 1 year	NA
Letter	HT-EDS	1 patient	Musculoskeletal pain, chronic pain and neuropathic pain	Case report
Article	Classic EDS Vascular EDS HT-EDS	1 patient 3 patients 20 patients	Musculoskeletal pain, neuropathic pain	Experimental study
Letter	Hypermobility syndrome *	12 patients	Nociceptive pain	Case-control
Article	Hypermobility syndrome *	29 patients	Musculoskeletal pain, joint pain	Case-control
Article	Classic EDS HT-EDS	4 patients 6 patients	NA	Case-control
Article	Osteoarthritis *	2243 patients	Knee pain	Prospective study
Review	HT-EDS	NA	Chronic pain, musculoskeletal pain	NA
Article	NA *	Mice	Neuropathic pain	Experimental study
Article	NA *	Rats	Neuropathic pain	Experimental study
Article	HT-EDS	27 patients	Neuropathic pain, chronic pain	Experimental study

The table shows the studies analyzing the characteristics of pain in various forms of EDS. The table also shows the most relevant data from the selected papers (study sample, study type, and form of EDS studied). Articles 31 and 36 from the bibliography are not included in the table because they did not meet the criteria for inclusion. * These works are relating to a pathology that has a symptomatology in common with EDS.

**Table 2 healthcare-11-00936-t002:** Studies referring to pain predators and pain management.

Publication	Classification of the Diseases	Patients Number	Pain Predictors	Pain Management	Type of Study
Article	HT-EDS	21 patients	SF-36 questionnaire	NA	Experimental study
Article	HT-EDS	32 patients	Health-related quality of life (HRQoL) RAND SF-36 questionnaire	NA	Case-control
Article	Hypermobility syndrome *	18 patients	SF-36 questionnaire	home-based exercise program	Experimental study
Article	HT-EDS	38 patients	HRQoL RAND SF-36 questionnaire	NA	Observational study
Article	EDS Marfan *	9 patients 15 patients	HRQoL SF-36 questionnaire	NA	Case-control
Letter	Classic EDS HT-EDS	9 patients 12 patients	Quality of life (QOL) SF-36 questionnaire	NA	Observational study
Article	EDS	250 patients	SF-36 questionnaire	NA	Observational study

The table Studies referring to pain predators and pain management are shown in the table. The table also shows the most relevant data from the selected papers (study sample, study type, and form of EDS studied). * These works are relating to a pathology that has a symptomatology in common with EDS.

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
