# Peer review of "Pain in Ehlers–Danlos Syndrome: A Non-Diagnostic Disabling Symptom?"

_healthcare, 2023, doi:10.3390/healthcare11070936_

Round 1

Reviewer 1 Report (Previous Reviewer 3)

Thank you for  the corrections and improvements you made!

Author Response

We thank the reviewer for appreciation regarding our work.

Reviewer 2 Report (New Reviewer)

Dear Authors,

The minor changes are shown in the annotated PDF document attached, and the major changes are listed below.

1. Explain why you used only two databases for the review and how you ensured the articles in the gray literature

2. Please include tables that describe the articles included in Sections 3.1 (Pain Characteristics), 3.2 (Predictors of Pain), and 3.3 (Management of Pain). The tables must contain all relevant details of the article and their quality assessment in the last column.

3. The discussion is very brief; please elaborate the discussion by explaining the most important finding in the articles included in the review. You may categorize the pain like you did in the results. Please provide your own interpretation.

Limitation: please provide based on the article quality assessment and your design limitation (databases)

Author Response

We would like to thank the reviewer for pointing out minor and major shortcomings in our work and also guiding us in improving our article.

Minor, point 2 and 3 changes, made based on advice received, are highlighted with yellow shading in the text.

Regarding the choice of databases and limitation, our goal was to make the information limitation explicit science-available for health care decisions with respect to the question posed. We queried the databases most commonly accessed by clinicians to explore the research question.The choice of literature from two sources could represent a limitation of sample constitution.  To minimize the Publication Bias, we will record the revision on the PROSPERO registry: this record aims to provide a comprehensive listing to avoid duplication and reduce opportunity for reporting bias by allowing for the comparison of the completed review with what was planned in the protocol.

Round 2

Reviewer 2 Report (New Reviewer)

Thanks a lot for addressing all my comments

This manuscript is a resubmission of an earlier submission. The following is a list of the peer review reports and author responses from that submission.

Round 1

Reviewer 1 Report

The author reviewed the correlation of pain with Ehlers-Danlos Syndrome based on the literature. As mentioned in the citation, there are several related reviews on this topic recently (ref.49, 50). It would be better if the authors acknowledge their works and point out the progress or new conclusion in this review.

Author Response

We appreciate the reviewer’s comment. The reported references are called narrative reviews of the literature by the authors. Our work is intended to be a systematic, hence reproducible, review of the literature. In this regard, we have revised the article to make these features clearer.

Reviewer 2 Report

The authors have conducted a study (the type of study is unknown) about an interesting topic which is apparently the relationship between Ehler-Danlos syndrome and pain. It seems that the authors have conducted a kind of review, however, there are some flaws that are a red flag to me. First, why the authors have not conducted a systematic review? Second, given the large number of topics that are covered in the article (pain prevalence, pain types, pain predictors, pain treatments, etc), it is not possible to understand the objective of the study. Authors say the have conducted a (non-systematic) review although it seems that it is more like a review topic. Even thought it could be a review topic, they still include excessive number of subtopics (as mentioned above).

Methods are short, unspecific, and clearly insufficient. The data is missing, as there is no extracted nor summarized data. It would not be possible to replicate this review. The statements and conclusions are not coherent. Furthermore, introduction, methods, results, discussion and conclusion are not aligned.

In summary, I am sorry to say that despite the interesting topic, the manuscript is unclear and not well structured.

Author Response

We thank the reviewer for pointing out some gaps in our work. To be more clear we have specified it is a systematic review and have made the primary and secondary objectives of the study more evident as highlighted in the revised paper.

We appreciate your insightful suggestions. Lines 93-95 show the strings as used with the goal of being as clear as possible and to make our research as reportable as possible. 

We thank the reviewer for the comment. We have modified the structure of paragraphs as rightly recommended to make our work more coherent. The changes we have made have been highlighted in the revised manuscript.

Reviewer 3 Report

Dear authors,

First of all, I would like to congratulate you for this interesting work and the innovative issue that you raise. It is a well written manuscript that complies with the international academic standards. 

I would suggest some quick and easy amendments and after that I am sure that it would be published. You can use the past tense to describe that something has already happened eg. The aim of this study was... You can also expand a little the discussion of the article using similar recent studies. Finally, take a quick look in the English language spelling and expressional improvements.

I wish you all the best and good luck!

Author Response

Thank you for the suggestion. We have expanded the discussion as highlighted in the revision of the article.

Round 2

Reviewer 1 Report

The author has made improvement for the review.

Reviewer 2 Report

The authors have indicated in the revised version that the type of study is a systematic review. It is good that this is now clarified, however, the major concerns of my first revision are still present and have not been adressed. These ones are related mainly to the methodology, which does not comply with basic standards for systematic review:

- The authors haven´t followed PRISMA guidelines.

- The review protocol haven´t been registered anywere.

- Criteria for study elegibility are defficient e.g., the type of studies included/excluded is missing (as it is the complete flowchart with the study selection process). How many disagreements did the researchers have?

- Outcome measures are poorly explained with no specific section.

- Research strategy is poor.

- How was the data extracted? which data did the authors extract from studies?

- How did the authors evaluate the quality of studies?

- Why there are no tables showing the data extracted and the quality of the studies evaluated?

In summary, this is not a systematic review.